# The Autophagy Nucleation Factor ATG9 Forms Nanoclusters with the HIV-1 Receptor DC-SIGN and Regulates Early Antiviral Autophagy in Human Dendritic Cells

**DOI:** 10.3390/ijms24109008

**Published:** 2023-05-19

**Authors:** Laure Papin, Martin Lehmann, Justine Lagisquet, Ghizlane Maarifi, Véronique Robert-Hebmann, Christophe Mariller, Yann Guerardel, Lucile Espert, Volker Haucke, Fabien P. Blanchet

**Affiliations:** 1Institut de Recherche en Infectiologie de Montpellier-IRIM-CNRS UMR9004, University of Montpellier, 34090 Montpellier, Francejustine.lagisquet@uk-erlangen.de (J.L.); ghizlane.maarifi@innovative-diagnostics.com (G.M.); veronique.hebmann@irim.cnrs.fr (V.R.-H.); lucile.espert@irim.cnrs.fr (L.E.); 2Leibniz-Forschungsinstitut für Molekulare Pharmakologie (FMP), Robert-Rössle-Straße 10, 13125 Berlin, Germany; mlehmann@fmp-berlin.de (M.L.); haucke@fmp-berlin.de (V.H.); 3Univ. Lille, CNRS, UMR 8576-UGSF-Unité de Glycobiologie Structurale et Fonctionnelle, F-59000 Lille, France; christophe.mariller@univ-lille.fr (C.M.); yann.guerardel@univ-lille.fr (Y.G.); 4Institute for Glyco-Core Research (iGCORE), Gifu University, Gifu 501-1112, Japan

**Keywords:** autophagy, dendritic cells, HIV-1, DC-SIGN, innate immunity

## Abstract

Dendritic cells (DC) are critical cellular mediators of host immunity, notably by expressing a broad panel of pattern recognition receptors. One of those receptors, the C-type lectin receptor DC-SIGN, was previously reported as a regulator of endo/lysosomal targeting through functional connections with the autophagy pathway. Here, we confirmed that DC-SIGN internalization intersects with LC3^+^ autophagy structures in primary human monocyte-derived dendritic cells (MoDC). DC-SIGN engagement promoted autophagy flux which coincided with the recruitment of ATG-related factors. As such, the autophagy initiation factor ATG9 was found to be associated with DC-SIGN very early upon receptor engagement and required for an optimal DC-SIGN-mediated autophagy flux. The autophagy flux activation upon DC-SIGN engagement was recapitulated using engineered DC-SIGN-expressing epithelial cells in which ATG9 association with the receptor was also confirmed. Finally, Stimulated emission depletion (STED) microscopy performed in primary human MoDC revealed DC-SIGN-dependent submembrane nanoclusters formed with ATG9, which was required to degrade incoming viruses and further limit DC-mediated transmission of HIV-1 infection to CD4^+^ T lymphocytes. Our study unveils a physical association between the Pattern Recognition Receptor DC-SIGN and essential components of the autophagy pathway contributing to early endocytic events and the host’s antiviral immune response.

## 1. Introduction

Due to their mucosal location and extensive network, dendritic cells (DC) are considered major sentinels of the immune system able to capture and present antigens to other immune cells, such as Ag-specific T cells, while also educating surrounding cells with exclusive cytokine secretion patterns [1,2,3,4]. DC are key antigen-presenting cells (APC) acting at the interface between innate and adaptive immunity due to the expression of different sets of pattern recognition receptors (PRR) and pathogen sensors enabling them to identify and respond to incoming microbes [5,6]. Interestingly, some PRR specifically expressed on DC subsets were shown to influence viral infection and transmission, notably regarding DC-mediated HIV-1 transfer toward CD4^+^ T lymphocytes [7,8].

Among these receptors, the C-type lectin receptor (CLR) dendritic cell-specific ICAM-grabbing non-integrin (DC-SIGN, also known as CD209 or CLEC4L) was characterized as a high-affinity receptor for mannosylated ligands. In particular, the mannose-capped lipoarabinomannan (ManLAM), a major cell wall component synthesized by the *Mycobacterium tuberculosis* complex, was reported to be a key DC-SIGN ligand able to influence anti-mycobacterial immunity [9,10]. Furthermore, the HIV-1 gp120 envelope glycoprotein and InterCellular Adhesion Molecule 3 (ICAM3 or CD50) were also shown to be high-affinity ligands for DC-SIGN [7,11]. While DC-SIGN was reported to enhance *trans*-infection of CD4^+^ lymphocytes [7,8,12], this CLR was also reported to be essential for the regulation of MHC-I- and MHC-II-mediated antigen presentation upon HIV-1 internalization [13,14,15,16] and for modulating TLR-driven innate immune responses [17,18]. The paradoxical impact of DC-SIGN during HIV-1 infection is not yet clear but might be explained by the functional dichotomy of this receptor, i.e., immune versus adhesion roles. As an important receptor involved in antigen presentation, DC-SIGN was connected to the endo/lysosomal compartment and acted early upon challenge with microbial ligands such as HIV-1, Dengue virus, or *Mycobacterium tuberculosis* [19,20], although some of these pathogens seem to timely interfere with lysosomal fusion events [9,21]. Interestingly, early events of DC-SIGN-mediated antigen uptake and intracellular trafficking were connected to autophagy, a major lysosomal degradation pathway [22,23,24].

Autophagy is a tightly regulated and evolutionary conserved lysosomal catabolic process implicated in many physiological functions, from development, stress regulation, and anti-inflammatory conditions to neurodegeneration [25,26,27]. The autophagy pathway is also recognized as a central component of the host’s innate and adaptive immune response against various intracellular pathogens [26]. Different forms of autophagy have been reported, and growing data in the literature suggest prominent specific regulations for autophagy-mediated lysosomal degradation of damaged organelles or antigens [28,29,30]. Autophagy is a multi-step cellular process controlled by autophagy-related proteins sequentially involved in vesicular initiation, elongation, and maturation [26,27]. This cellular pathway is, therefore, highly dynamic and translated as a regulated flux in which the lipidated autophagosomal marker LC3 (LC3-II) can be biochemically monitored to assess autophagy status [30]. The autophagy initiation step is regulated by multiprotein complexes, among which the ATG9/WIPI2/ATG2 complex appears essential to supply membranes required for autophagy-related vesicle formation [31,32]. The autophagy factor ATG9 is a multi-membrane-spanning protein integrated into the outer membrane of autophagosomes and essential for autophagy initiation [33,34,35].

Here, we identified the autophagy initiation factor ATG9 complexed to the CLR DC-SIGN upon ligand-driven endocytosis. Engagement of DC-SIGN promotes autophagy flux activation and coincides with an interconnection between the endocytosed CLR DC-SIGN and LC3^+^/ATG9^+^ intracellular structures related to a non-canonical autophagy pathway. Super-resolution imaging analyses evidenced the formation of intracellular ATG9-containing nanoclusters upon DC-SIGN engagement, suggesting a local and specific induction of autophagy-related organelles. Finally, we demonstrate the critical role of ATG9 in DC-SIGN-mediated autophagy flux activation and in contributing to dampening both HIV-1 capture in human dendritic cells and DC-mediated transfer of infection of CD4^+^ T lymphocytes. Our study sheds light on a novel autophagy-associated endocytic pathway engaged concomitantly to autophagy flux induction upon DC-SIGN-associated ATG9 complex that could rapidly modulate antiviral host response.

## 2. Results

### 2.1. Ligand-Induced DC-SIGN Endocytosis

Human primary monocyte-derived dendritic cells (MoDC) were used and treated with ligands previously reported to drive DC-SIGN endocytosis. Flow cytometry analyses of MoDC revealed that 30% to 50% of cell surface DC-SIGN were endocytosed upon 20 to 40 min of challenge with DC-SIGN-specific antibodies (Histograms in Figure 1A), Mycobacterium-derived mannose-capped lipoarabinomannan (ManLAM) (Histograms in Figure 1B) or even Human Immunodeficiency Virus-1 (HIV-1) (Histograms in Figure 1C). Immunofluorescence analyses confirmed the endocytic pattern of DC-SIGN upon treatment of MoDC with all these ligands (Figure 1A–C, Imaging insets). All tested ligands on MoDC were thus able to induce DC-SIGN internalization. As expected from the literature, we validated that our DC-SIGN endocytosis model occurred mainly via a clathrin-dependent mechanism (Figure 1D) [13,14]. Indeed, more than 60% of DC-SIGN signal co-localized with clathrin^+^ vesicles upon 15 min of receptor engagement with DC-SIGN antibodies (Figure 1D).

### 2.2. Engaged DC-SIGN Co-Precipitates with LC3 and Leads to Autophagy Flux Induction

DC-SIGN endocytosis was reported to intersect with intracellular autophagy-related vesicles. Indeed, a physical association between DC-SIGN and LC3 could be confirmed by a biochemical approach in which constructs expressing different GST-LC3 isoforms were separately co-transfected in HEK293T cells with a construct expressing wild-type DC-SIGN. Cells were then incubated with DC-SIGN ligands on ice (time 0) or at 37 °C for 30 min, and the corresponding lysates (Figure 2A, left) were submitted to GST-pull-down (Figure 2A, right).

These experiments confirmed that DC-SIGN can be pull-downed with different ectopic LC3 isoforms, and this interaction appeared markedly increased upon receptor engagement (Figure 2A, right). Although DC-SIGN seemed to be preferentially associated with the LC3A isoform, all LC3 isoforms could interact with the receptor (Figure 2A). The association between endocytosed DC-SIGN and LC3^+^ structures appeared to be also functionally related, as evidenced by an increase in the autophagosomal form of LC3 (LC3-II) observed very early upon treatment of MoDC with anti-DC-SIGN (Figure 2B, left). Furthermore, MoDC pre-treated with chloroquine, a lysosomotropic agent interfering with lysosome-mediated degradation, displayed much higher LC3-II intracellular accumulation upon DC-SIGN engagement, demonstrating a specific DC-SIGN-dependent autophagy flux induction (Figure 2B). The same DC-SIGN-dependent autophagy activation pattern was observed when MoDC were pre-treated with bafilomycin A1, a macrolide specifically targeting the vacuolar-type H^+^-ATPase (V-ATPase) (Appendix A). The DC-SIGN-mediated autophagy flux induction was specific as further controlled with unrelated isotype-matched Abs, which did not significantly promote LC3-II induction (Figure 2B, right). Importantly, an increase in the LC3 autophagosomal form was also observed in chloroquine-treated MoDC incubated with natural DC-SIGN ligands, such as ManLAM (Figure 2C) or HIV-1 (Figure 2D). Collectively, our data suggest that DC-SIGN engagement regulates autophagy flux induction and intersection with autophagy-related endosomal structures in MoDC.

### 2.3. Endocytic Screening Identifies Exclusive DC-SIGN-Associated Autophagy-Related Proteins

DC-SIGN endocytosis was shown to traffic through the endo-lysosomal degradation pathway, thus regulating antigen processing and antigen presentation toward Ag-specific CD4^+^ T lymphocytes. We and others have shown that DC-SIGN endocytosis also intersected with the autophagy pathway [22,23,24], suggesting a link between both processes. We, therefore, performed an immunofluorescent microscopy screening to investigate co-localization events between internalized DC-SIGN and selected autophagy-related factors in MoDC (Figure 3). As expected, DC-SIGN internalization intersected with intracellular punctate LC3^+^ structures as early as 15 min (Figure 3A), validating our read-out. Co-localization between LC3 and endocytosed DC-SIGN was even enhanced after 30 min of receptor engagement (Figure 3A and Histograms in Figure 3B). Interestingly, we often observed that DC-SIGN/LC3 co-localizing signals appeared close to the plasma membrane early upon receptor engagement (Figure 3A, left insets). Our non-exhaustive screening assay also uncovered an association between DC-SIGN and the autophagy vesicle nucleation factor ATG9a (hereafter named ATG9) upon antibody-mediated receptor internalization (Figure 3A). Other autophagy initiation (ULK-1, ATG13) or elongation (ATG16L1) factors, as well as autophagy receptors (such as p62/SQSTM1), were seemingly not associated with DC-SIGN (Figure 3A). Quantification of intensity correlation co-localizing DC-SIGN/ATG9 signals revealed that internalized DC-SIGN was closely and significantly associated with ATG9 very early upon engagement of the receptor with anti-DC-SIGN. However, the major endocytic pathway remained clathrin-dependent (Figure 1D and Figure 3B).

### 2.4. DC-SIGN Ligands Promote Receptor Association with ATG9^+^ Structures

Co-localization between ATG9 and endocytosed DC-SIGN was further confirmed and quantified in MoDC treated with specific DC-SIGN antibodies (Figure 4A), and quantifications of intensity correlation co-localizing DC-SIGN/ATG9 signals confirmed a significant association between internalized DC-SIGN and ATG9 very early upon engagement of the receptor with anti-DC-SIGN (Figure 4A, Histogram). Interestingly an even slightly enhanced similar pattern was observed upon engagement of DC-SIGN with ManLAM (Figure 4B) and HIV-1 (Figure 4C).

Surprisingly, although ATG9 intracellular profile was mainly reminiscent of the endoplasmic reticulum (ER)/Golgi pattern, internalized DC-SIGN was preferentially showing up with sub-membrane endosomal ATG9^+^ vesicles early on upon endocytosis of the receptor in MoDC (Figure 4 and arrows in Appendix A). Such feature could be explained by internalized receptors intersecting with plasma membrane-associated ATG9 or even suggest repolarization of intracellular organelles, such as the ER-Golgi, toward sites of DC-SIGN engagement at the plasma membrane. Kinetic parameters of the above results, combined with our autophagy flux induction data, suggested that ATG9 contributed to the formation of autophagosomal structures linked to endocytosis (amphisomes) upon DC-SIGN engagement. Interestingly, we observed that MoDC pre-treated with chloroquine and a potent autophagy flux inducer, such as Torin1, showed highly prominent and enlarged DC-SIGN^+^/ATG9^+^ intracellular vesicles localized close to the plasma membrane early upon receptor engagement (arrowheads in immunofluorescence lower panels of Appendix A). Consistently, a rapid and specific appearance of an autophagosomal lipidated form of LC3 (LC3-II) in parallel to a decreased signal in the autophagic receptor p62/SQSTM1 could be observed by Western blotting upon receptor engagement (Appendix A). The addition of chloroquine impeded p62/SQSTM-1 degradation, as expected, suggesting a functional autophagy flux in MoDC pharmacologically treated and incubated with DC-SIGN ligands (Appendix A). Our data, therefore, suggested that drug-mediated autophagy flux induction can fuel and preset a potential non-canonical ATG9-dependent DC-SIGN autophagy pathway.

### 2.5. Physical Association between Internalized DC-SIGN and ATG9

We analyzed the physical association between DC-SIGN and ATG9 using an engineered cell line stably expressing DC-SIGN (HEK-DC-SIGN) and transfected with plasmid constructs expressing HA-ATG9 (Appendix A). HEK-DC-SIGN cells displayed intracellular and surface DC-SIGN expression at levels similar to those usually observed in MoDC (Appendix A). When pre-treated with bafilomycin A1 and incubated for 60 min with anti-DC-SIGN, HEK-DC-SIGN cells evidenced increased levels of LC3-II, as shown in Western blotting data on extracted cell lysates (Appendix A), therefore confirming the functionality of DC-SIGN-mediated autophagy flux induction in these cells. We then treated or not HEK-DC-SIGN cells, previously transfected with a control plasmid (ø) or encoding HA-ATG9, with the DC-SIGN ligand ManLAM for 30 min and subjected cell lysates to anti-HA immunoprecipitation. Our results showed that DC-SIGN co-immunoprecipitated with HA-ATG9, with levels slightly increased upon receptor engagement (Appendix A). Interestingly, a topological HA-ATG9 mutant (HA-ATG9 N99D) was still found to co-immunoprecipitate DC-SIGN to levels similar to wild-type HA-ATG9 (Appendix A). Our data, therefore, confirmed a physical association between ATG9 and DC-SIGN, even in engineered cell line models.

### 2.6. ATG9 Forms Intracellular Nanoclusters with Endocytosed DC-SIGN and Is Required for DC-SIGN-Mediated Autophagy Flux Induction

To further characterize the observed DC-SIGN^+^/ATG9^+^ amphisomes, we turned to super-resolution imaging of the vesicular structures formed early upon DC-SIGN engagement and internalization on MoDC. We first validated the sub-membrane assembly of endocytosed DC-SIGN with ATG9-containing membranes by confocal microscopy upon 30 min of receptor engagement compared to control cells (Figure 5A, left panels). Within the DC-SIGN^+^/ATG9^+^ structures, STimulated Emission Depletion (STED) microscopy with a resolution of <50 nm clearly revealed polarized ATG9-containing domains clustering and shrouding endocytosed DC-SIGN receptors and forming large intracellular vesicles upon 30 min of anti-DC-SIGN engagement (Figure 5A, right panels).

The same ligand-mediated receptor endocytosis kinetic assay as before was performed and showed that large intracellular vesicles were formed as early as 15 min upon DC-SIGN engagement (Figure 5B). Clustering and assembly of internalized DC-SIGN in larger endosomal vesicular structures were also evidenced and confirmed upon analysis of receptor localization with a fire-scale pattern (Figure 5C). Analysis of DC-SIGN^+^ intracellular vesicles revealed an increased frequency of enlarged vesicular structures as early as 15 min of DC-SIGN engagement with an average diameter of 500 nm compared to control conditions (0 min) (Figure 5D). Our data suggest the formation of ATG9-bearing nanostructured membranes rapidly polarized towards sites of DC-SIGN receptor engagement.

### 2.7. ATG9 Is Required for DC-SIGN-Mediated Autophagy Flux Induction

To further assess ATG9 function in DC-SIGN-mediated autophagy activation, we transfected primary human MoDC with irrelevant control siRNA (siCtrl-MoDC) or ATG9-specific siRNA (siATG9-MoDC) to efficiently downregulate ATG9 expression (~70 downregulation as shown in Figure 6, left graph). Cells were then pre-treated or not with bafilomycin A1 before incubation or not with ManLAM (Figure 6) or DC-SIGN-specific antibodies (Appendix A). As expected, LC3-II was increased in bafilomycin A1-treated siCtrl-MoDC cells upon DC-SIGN engagement, confirming DC-SIGN-mediated autophagy flux activation (Figure 6).

However, autophagy flux was markedly compromised in siATG9-moDC as evidenced by the decreased levels of LC3-II in basal and stimulated conditions upon bafilomycin A1 pre-treatment (Figure 6 and Appendix A). Our results not only showed a critical contribution of ATG9 in basal cellular autophagy flux but also confirmed its involvement in DC-SIGN-mediated autophagy flux induction.

### 2.8. ATG9 Expression Limits HIV-1 Infection of MoDC and Transmission of Infection to CD4^+^ T Cells

The link between the ATG9-associated DC-SIGN complex and autophagy-mediated lysosomal degradation made us wonder if this pathway could contribute to an autophagy-dependent antiviral response previously reported in MoDC against pathogens such as HIV-1. MoDC were, therefore, left untransfected (MoDC-NT) or transfected with irrelevant siRNA (MoDC-siCtrl) or siRNA directed against ATG9 (MoDC-siATG9). Cells were then left uninfected (MoDC NI) or challenged with R5-tropic HIV-1 wild-type for a short time point (8 h) before analyzing the anti-HIV1-Gag signal by flow cytometry (Figure 7A). This kinetic viral capture assay was already shown to be independent of viral reverse transcriptase activity [22]. We could observe a 3- to 4-fold increase in HIV-1-Gag signal in siATG9-MoDC when compared to NT- and siCtrl-MoDC (Figure 7A), suggesting the existence of an ATG9-dependent cellular mechanism able to limit or lower HIV-1 entry into myeloid DC, reminiscent of the phenotype obtained upon downregulation of other critical autophagy-related cellular factors [22]. As such, a significant increase in viral input was measured in HIV-1-R5 challenged siATG9-MoDC generated from 4 different donors (Figure 7A, right graph). We then co-cultured HIV-1-R5-infected siRNA-treated MoDC with autologous CD4^+^ T lymphocytes in order to analyze the transmission of infection. Our data showed that infected siATG9-MoDC were two to three times more prone to transmit HIV-1 infection towards CD4^+^ T cells compared to HIV-1-R5-infected NT- and siCtrl-MoDC (Figure 7B). These data, therefore, reveal the existence of an antiviral ATG9-associated CLR complex induced upon ligand binding to DC-SIGN and acting very early upon the HIV-1 challenge. The formation of this complex close to the plasma membrane relies on the membrane nucleating activity of the essential autophagy-related factor ATG9 which might regulate C-type lectin receptor post-endocytic sorting toward an autophagy-mediated lysosomal compartment contributing to an early antiviral cellular response (Figure 8).

## 3. Discussion

In this study, we revealed the formation of a novel protein complex linked to the pathogen recognition receptor DC-SIGN and the autophagy-related proteins ATG9, known to regulate autophagy initiation. We and others reported that DC-SIGN post-endocytic sorting could intersect with autophagy-related intracellular vesicles. These organelles issued from a merge between endosomal and autophagosomal vesicles (amphisomes) were named immunoamphisomes due to their ability to regulate innate and adaptive immune responses [22,23]. In addition to the confirmed association between endocytosed DC-SIGN and LC3^+^ vesicles, our current study further demonstrated that all the DC-SIGN ligands used also induced a rapid and specific cellular autophagic response in MoDC.

We hypothesized that DC-SIGN engagement could be linked to an early autophagic response with pre-autophagy-related structures formed very early upon engagement of the CLR. In agreement with this hypothesis, we observed a significant co-localization between endocytosed DC-SIGN and ATG9^+^ vesicles localized closely to the plasma membrane upon engagement of the receptor. Noteworthy, some ATG-related factors were functionally connected to the endosomal pathway, seemingly supplying membranes toward the formation of autophagosomal structures. Accordingly, ATG9 and ATG16L1 proteins were both reported to contribute to the formation of pre-autophagosomal structures with membrane sources from endosomes and cellular plasma membranes [34,35,36,37]. Interestingly, intracellular ATG9^+^ vesicles were shown to represent a significant membrane source for autophagosome formation [34,35,38] while seemingly originating from the Golgi apparatus, plasma membrane, and endosomal compartments. However, the main autophagosome initiation site within cells has been reported to be localized at the ER with defined structures called omegasomes [39]. Our data showed the proximity between DC-SIGN^+^/ATG9^+^ vesicles and the plasma membrane, suggesting a local and rapid initiation of autophagy-related organelles transitioning to LC3^+^ immunoamphisomes upon receptor engagement. The association with ATG9^+^ isolation membranes seems, therefore, to initiate a very early trafficking fate of incoming virions towards the endo-lysosomal degradation pathway as previously suggested [15,16,40,41,42]. Interestingly, close appositions between the plasma membrane and the ER were described, and such membrane contact sites were also reported to critically regulate ions homeostasis, signaling events, as well as phosphoinositides and lipid metabolism [43,44,45]. Furthermore, ER-plasma membrane contact sites were shown to regulate autophagosome biogenesis through the implication of extended synaptotagmins (E-Syts) [39,43,44,46]. One plausible hypothesis could be that such contact sites would be favored in conditions of receptor engagement and cell cytoskeleton polarization allowing for a quick and localized supply of lipids and cellular factors necessary to generate specific autophagy-related organelles associated with endocytosis. Accordingly, our super-resolution microscopy approach clearly evidenced the appearance of larger submembrane clusters containing ATG9 upon receptor engagement. A better ultrastructural and biochemical study on those submembrane ATG9^+^/DC-SIGN^+^ clusters will be required to state if other components of autophagy-initiation complex, phosphatidylinositol synthase (PIS)-enriched ER subdomains and ATG9 vesicles can act in concert with DC-SIGN-containing endosomes to generate immunoamphisomes.

While other major ATG-related proteins (such as ULK1) were not significantly found to be associated with internalized DC-SIGN in our model, we could not totally exclude their implication in membrane nucleation and post-endocytic sorting events upon DC-SIGN engagement. Hence, we observed that mTOR inhibitors, and promoters of canonical autophagy, can preset ATG9^+^ vesicles which appear enlarged upon DC-SIGN engagement. This is not totally unexpected as ATG9^+^ vesicles required for membrane expansion were shown to be recruited to autophagy initiation sites through their interaction with the scaffolding protein ATG17 (RB1CC1/FIP200 is the mammalian orthologue) while ATG1 kinase (ULK1/2 orthologue)-mediated ATG9 phosphorylation was also reported to control both the localization to the pre-autophagosomal structure (PAS) and the rate of autophagosome formation [33,38,47,48]. A better characterization of the protein complexes associated with ATG9^+^ vesicles and the analysis of ATG9 post-translational modification status at the plasma membrane will be required to further delineate cellular machinery linking ATG9^+^ vesicles with the endosomal pathway and autophagy vesicles nucleation. Noteworthy, our data with an ATG9 mutant (ATG9-N99D) topologically altered in membranes still showed association with DC-SIGN, suggesting that membrane topology of ATG9 is not required for membrane expansion in agreement with previous studies [33].

The rapid association between DC-SIGN and ATG9^+^ vesicles upon receptor engagement and the intersection toward LC3^+^ intracellular vesicles also raised the question of the endocytic route engaged upon ligand-mediated receptor binding and internalization. Our data suggested that DC-SIGN post-endocytic sorting and routing could be functionally linked to a non-canonical autophagy pathway. Non-canonical forms of autophagy are often characterized by the conjugation of ATG8 to single membranes (CASM) [49], which depending on the lipidation state of LC3, contribute to endolysosomal and endophagosomal trafficking events such as LC3-associated phagocytosis (LAP) or LC3-associated endocytosis (LANDO) [50,51,52,53]. Importantly, these processes were shown to rely on a subset of core autophagy-related factors usually involved in canonical autophagy [51,52]. Of importance, LAP was reported to favor phagosomal maturation while also prolonging MHC-II antigen processing [54] and regulating DC immune functions [55,56]. Interestingly, the LAP outcome was therefore sharing some phenotypic and functional features obtained upon DC-SIGN engagement and could represent one of the pathways activated downstream of ligand-induced DC-SIGN endocytosis. Although DC-SIGN has never been shown to contribute to LAP, other closely relative CLR were reported to regulate this cellular process critical for killing certain pathogens, such as fungi, parasites, or helminths [50,56]. For example, the CLR dectin-1 (CLEC7A), a β-glucan receptor, was required to recruit phagosomal LC3 and facilitate MHC-II presentation of fungal-derived antigens [57]. Furthermore, the immunomodulatory functions of dectin-2 (CLEC6A), a high-mannan binding receptor-like DC-SIGN mainly expressed on macrophages, myeloid DC, and plasmacytoid DC, were shown to rely on the induction of LAP in a mouse model of Bone marrow-derived DC challenged with commensal non-pathogenic yeast [55]. Because dectin-2 could share signal pathways similar to dectin-1 and its intracellular trafficking and connections to the endo/lysosomal compartment would behave quite similarly to DC-SIGN, the involvement of LAP downstream of DC-SIGN engagement could be highly plausible. More investigation will be therefore required to assess the contribution, if any, of LAP downstream of DC-SIGN. LAP was also shown to require cytosolic NADPH oxidase factors (NOX) in order to generate reactive oxygen species (ROS) necessary for LC3-II formation and recruitment to the phagosomal membrane before fusion between the newly created LAPosomes and lysosomes [50,58]. Although some CLR, such as dectin-1, MR, or dectin-2, were reported to produce ROS upon ligand binding, DC-SIGN was rather shown to act in a ROS-independent manner, making this CLR unlikely to regulate LAP [59]. Interestingly, another non-canonical autophagy pathway connected to clathrin-mediated endosomal structures was recently reported as essential for mounting specific immune responses. Indeed, LANDO in microglia was shown to involve the conjugation of the autophagy protein LC3 to Rab5^+^/clathrin^+^ endosomes and appeared critical to regulating immune-mediated aggregate removal and microglial activation in a murine model of Alzheimer’s Disease [52]. Noteworthy, while conditions of autophagy induction in different cell types were shown to limit infection against HIV-1 [22,60,61] as well as other RNA (CHIKV, SINV, WNV, RSV) [60,62,63] or DNA viruses (HSV-1, HCMV) [64,65,66] non-canonical forms of autophagy and some ATG proteins were recently reported to be interconnected with essential innate immune factors and pathways [67,68].

Our data suggest the involvement of a non-canonical endosomal autophagy pathway similar to previous reports [34,35,36] but dependent on a major innate immune receptor and possibly linked to host response against pathogenic challenge. Future studies will determine which kind of regulation ATG9 imposes on DC-SIGN signaling. Indeed, ATG9 was reported as a negative or positive regulator of the STING/TBK1 or TRAF6/JNK innate immune signaling pathways, respectively [69,70]. In future studies, it will be worth examining the role of the ATG9/DC-SIGN complex in innate immune responses to viral challenges and the consequences on host adaptive immunity, a major immune arm regulated by both autophagy and DC-SIGN.

## 4. Materials and Methods

### 4.1. Cells

Human monocytes from buffy coats were obtained according to institutional guidelines of the ethical committee of the CNRS and EFS. After the Ficoll gradient on PBMC, monocytes were isolated using CD14 MicroBeads (Miltenyi Biotec, Bergisch Gladbach, Germany). Usual purity was >95% CD14^+^. Human MoDC were generated by incubating purified monocytes in IMDM supplemented with 10% FCS, 2 mM L-glutamine, 100 IU/mL penicillin, 100 μg/mL streptomycin, 10 mM Hepes, 1% non-essential amino acids, 1 mM sodium pyruvate and cytokines GM-CSF (500 IU/mL) and IL-4 (500 IU/mL) (both from Miltenyi Biotec, Bergisch Gladbach, Germany). The obtained immature MoDC were harvested at days 5–6 and phenotyped by flow cytometry before experimental use. The adherent cell lines HEK293T and HEK-DC-SIGN (engineered in this study) were maintained in supplemented DMEM under puromycin selection for the latter only.

### 4.2. Virus and Lentiviral Vectors Production

Virus stocks were produced as previously described [71,72]. Briefly, plasmid pR8-Bal, encoding for full-length HIV-1 R5 strain provirus, was transiently transfected in HEK293T producer cells by the calcium-phosphate co-precipitation method. Infectious titers of viral stocks were evaluated by limiting dilution on HeLa TZM-bl cells or JLTR-GFP lymphoid cells. Viral titers were expressed as infectious units (IU) per ml, and 500 ng of p24^gag^ on 2.5 × 10^5^ CD4^+^ HeLa P4-R5 corresponds to an MOI of 1 (multiplicities of infection). Viral physical titers were evaluated by quantification of HIV-1 p24^gag^ by ELISA using Innotest^®^HIV antigen mAb (Fujirebio, Tokyo, Japan).

### 4.3. Plasmids

The following reagent was obtained through the NIH AIDS Reagent Program, NIAID, NIH: pcDNA3-DC-SIGN from Drs. S. Pöhlmann, F. Baribaud, F. Kirchhoff and R.W. Doms [73]. DC-SIGN gene was subcloned in the lentiviral vector system pRRL.sin.cPPT.SFFV/IRES-puromycinR.WPRE used to generate the cell lines stably expressing the genes of interest has been described previously. Expression plasmid encoding HA-ATG9 was a kind gift from Dr. Sharon Tooze. GST-LC3 constructs were generated using the Gateway^TM^ cloning strategy using pDEST15 as destination vectors, as already reported [74]. All plasmids were transfected in indicated cell lines using the transfection reagents jetPEI^®^ (Polyplus-transfection, Illkirch-Graffenstaden, France) or TurboFect (Thermo Fisher Scientific, Waltham, MA, USA) according to the manufacturer’s instructions.

### 4.4. Chemicals and Reagents

All chemicals were obtained from Sigma unless stated otherwise. For autophagy flux experiments, Torin 1 (Enzo life sciences, NY, USA) and chloroquine were used at 1 μM and 50 μM, respectively. The Protein A/G sepharose reagent used for immunoprecipitation experiments was from Rockland Immunochemicals (Rockland, NY, USA). Mannose-capped lipoarabinomannan was purified from *Mycobacterium bovis* BCG, as previously reported [75,76]. Briefly, cells were harvested, washed in PBS, and disrupted by French press in 8% (*v*/*v*) Triton X-114, 5 mM EDTA, and 10 mM MgCl_2_. After phase separation at 37 °C, lipoglycans were collected in the lower phase, precipitated by cold ethanol, and extracted by saturated phenol. ManLAM was then separated from other lipoglycans (lipomannan and phospho-inositol mannosides) by gel filtration on a Sephacryl S-200 column (GE Healthcare, Amersham, UK) irrigated with tris deoxycholate buffer (10 mM Tris-HCl pH 8.0, 10 mM EDTA, 0.2 M NaCl, 0.25% deoxycholate). The purity of the eluted fractions was assessed by 13% SDS-PAGE and staining for carbohydrates. The structural integrity of the lipoglycan was checked by multi-nuclear magnetic resonance spectroscopy as previously described [76]. For immunofluorescence assays, DAPI (4′,6′ Di Amidino-2-Phényl Indole) and Prolong^®^Gold were obtained from Thermo Fisher Scientific (Waltham, MA, USA).

### 4.5. Antibodies

DC-SIGN engagement and internalization assays were performed using monoclonal anti-DC-SIGN antibodies (clone MR1 from Santa Cruz Biotechnology (Dallas, TX, USA) and clone DCN46 from BD Biosciences (Franklin Lakes, NJ, USA)). Other anti-DC-SIGN Abs were used for flow cytometry (clone eB-h209 from ebioscience; #REA617 from Miltenyi Biotec (Bergisch Gladbach, Germany) and #9E9A8 from Biolegend (San Diego, CA, USA) and Western blotting (#D7F5C from Cell Signaling Technology, Danvers, MA, USA). Additionally, the following reagents were obtained through the NIH AIDS Reagent Program, Division of AIDS, NIAID, NIH and used for flow cytometry or Western blotting applications: Anti-Human DC-SIGN/L-SIGN Monoclonal (14E3G7) from Dr. Ralph Steinman; Anti-Human DC-SIGN/DC-SIGNR Monoclonal (DC28) from Drs. F. Baribaud, S. Pöhlmann, J.A. Hoxie and R.W. Doms. Antibodies against P62/SQSTM-1 (#H290) were from Santa Cruz Biotechnology (Dallas, TX, USA). Anti-LC3B (#L7543), anti-GAPDH-HRP (#G9295), anti-ATG13 (#SAB1307145), anti-HA (#H6908) and anti-GST-HRP (#A7340) were from Sigma-Aldrich (Merck KGaA, Darmstadt, Germany). Antibodies against ATG16L1 (ab187671), ULK1 (ab128859), ATG9 (ab108338), Lamp1 (ab24170), anti-P62/SQSTM-1 (ab56416) and HA-tag (ab9110) were from Abcam (Cambridge, UK). Anti-LC3 (PM036) and anti-ATG16L (PM040B) were obtained from MBL International (Woburn, MA, USA), while monoclonal anti-actin (clone C4) was from Millipore (Merck KGaA, Darmstadt, Germany). When not stated otherwise, all FACS antibodies used for primary human cell phenotyping were obtained from BD Biosciences (Franklin Lakes, NJ, USA) or Miltenyi Biotec (Bergisch Gladbach, Germany). All coupled secondary antibodies used for Western blotting and immunofluorescence assays were purchased from Sigma-Aldrich (Merck KGaA, Darmstadt, Germany) and Thermo Fisher Scientific (Waltham, MA, USA), respectively. Rabbit polyclonal anti-Clathrin antibodies were a kind gift from Paul Mangeat (Montpellier, France).

### 4.6. Western Blotting

Cells were lysed for 20 min at 4 °C in lysis buffer (20 mM Tris pH 7.5, 150 mM NaCl, 1% NP40, 1 mM NaVO_4_, 50 mM NaF, 10 mM Na_4_P_2_O_7_, protease inhibitor cocktail (Roche, Bâle, Switzerland)) and centrifuged at 26,000× *g* for 20 min. Post-nuclear Supernatants were submitted to electrophoresis on home-made 10% Prosieve (Lonza, Bâle, Switzerland) gels or Bolt 4–12% Bis-Tris Plus pre-cast gels before transfer to a nitrocellulose membrane (GE Healthcare, Amersham, UK) and then immunoblotted as indicated. Membranes were stripped in 0.1 M Glycine buffer (PH 2.3) in case of re-blotting and loading control. Membranes were revealed with Clarity^TM^ Western ECL substrate (Bio-Rad Laboratories, Hercules, CA, USA). Chemiluminescence was acquired (Chemidoc, Bio-Rad Laboratories, Hercules, CA, USA), followed by densitometry analysis (Image Lab^TM^ software; Bio-Rad Laboratories, Hercules, CA, USA).

### 4.7. Immunoprecipitation Assays

For immunoprecipitation assays, 5 × 10^6^ HEK293T cells were co-transfected with indicated encoding plasmids and harvested 36–48 h post-transfection. Lysates were prepared as described for the Western blotting method and incubated overnight on Protein A/G sepharose beads pre-coated with anti-HA antibodies. In most immunoprecipitation assays performed, lysates were pre-cleared on uncoupled Protein A/G sepharose beads before immunoprecipitation. The resulting immunoprecipitates were washed three times with lysis buffer and resuspended in Laemmli denaturing sample buffer, followed by Western blot analysis.

### 4.8. GST Pull-Down Experiments

HEK293T cells in 6-well plates were co-transfected with plasmids expressing DC-SIGN WT and either expressing GST-fusion proteins or GST alone. Cell lysis was performed 48h post-transfection, and lysates were incubated with glutathione-coupled sepharose beads (GE Healthcare, Amersham, UK) for 3 h at 4 °C. Unbound proteins were removed by washings. Bound proteins were then eluted by boiling for 10 min in SDS gel loading buffer and separated by gel electrophoresis.

### 4.9. siRNA Transfections

SMARTpool: ON-Target Plus human siRNA directed against ATG9a (#L-014294-01-0005) as well as siCtrl (5′-AAATGAACGTGAATTGCTCAA-3); siRNA sequence specific for Luciferase) were obtained from Dharmacon (Lafayette, CO, USA). HiPerFect Transfection Reagent (Qiagen, Hilden, Germany) or INTERFERin^®^ (Polyplus-transfection, Illkirch-Graffenstaden, France) were used for siRNA transfection according to the manufacturer’s recommendations and as previously reported [20]. Briefly, 4 × 10^5^ iDC were usually transfected with 100 nM siRNA in 500 μL of IMDM/1% SVF medium in 12-well plates. A second round of transfection was performed 24 h later. Specific gene knockdowns were controlled by Western blotting compared with untransfected and siCtrl conditions and normalized to loading control.

### 4.10. Infections

For immunofluorescence assays, when not stated, HIV-1 treatment was performed on cells previously plated on glass coverslips at 2.5 μg p24 HIV-1 for 2.5 × 10^5^ cells (corresponding to an MOI of 5) and adapted depending on cell number. Functional infection assays and viral capture were conducted in 96-well plates in which 2.5 × 10^5^ cells were challenged with 1 μg p24 HIV-1 (corresponding to an MOI of 2) for 6–8 h. Cells were analyzed by flow cytometry upon fixation and anti-HIV-1-Gag staining. HIV-1 challenged MoDC were also co-cultured with autologous CD4^+^ T cells (ratio 1:1) for 72 h before the fixation and analysis of HIV-1-Gag signal in CD3^+^ cells.

### 4.11. Flow Cytometry Assays

Surface staining for phenotyping of primary cells (monocytes, MoDC, and autologous CD4^+^ T cells when required) was performed in FACS buffer (PBS/1% bovine serum albumin) at 4 °C for 30 min using mAbs from BD Biosciences (Franklin Lakes, NJ, USA) and Miltenyi Biotec (Bergisch Gladbach, Germany) and directed against the following molecules: CD1a, CD3, CD4, CD8, CD14, CD45Ro, CD69, CD83, CD86, DC-SIGN, HLA-DR. For DC-SIGN internalization assays, a cocktail of mouse monoclonal antibodies (DCN46 (BD) and MR1 (Santa Cruz Biotechnology, Dallas, TX, USA) at 5 μg/mL each) or ManLAM (2 μg/mL) were incubated for 30 min on cells previously stored at 4 °C. Internalization times were starting when cells were switched back to 37 °C for indicated kinetic times. The time 0 corresponds to cells left with reagents at 4 °C as above but processed straight without incubation at 37 °C. DC-SIGN levels in antibody-mediated internalization were measured upon staining with only anti-mouse fluorescently labeled secondary antibodies. For ManLAM-mediated DC-SIGN internalization, receptor levels were analyzed by flow cytometry upon staining with anti-DC-SIGN-APC (Biolegend) or anti-DC-SIGN-PE (Miltenyi Biotec, Bergisch Gladbach, Germany). Isotype-matched mAbs were used in all experiments as negative controls. Samples were analyzed by flow cytometry using a FacsCalibur (Becton Dickinson, Franklin Lakes, NJ, USA) and data processing with FlowJo^®^software (FlowJo, Ashland, OR, USA).

### 4.12. Immunofluorescence Assays

For immunofluorescence, cells were left to adhere on poly-L-lysine treated glass coverslips for 30–60 min at 37 °C. Upon experimental use, cells were then fixed, permeabilized, and stained with indicated antibodies, followed by fluorescently labeled secondary antibodies from Molecular probes (Thermo Fisher Scientific, Waltham, MA, USA). Preliminary epifluorescence acquisition was performed using a Leica microscope, and samples were analyzed using the LAS software (Leica Camera, Wetzlar, Germany). Confocal microscopy acquisitions were conducted on a Zeiss LSM780, and data analyses were performed using ZEN software (Zeiss, Oberkochen, Germany). When required, Z-series of optical sections were performed at 0.5 μm increments for qualitative analysis. Green, red, and far red fluorescences were acquired sequentially. Colocalization events were calculated using intensity-based Manders’ M coefficient obtained from the JACoP plug-in with ImageJ Software (NIH, Bethesda, MD, USA), and statistical quantifications were performed with GraphPad Prism (GraphPad Software, San Diego, CA, USA).

### 4.13. Gated Stimulated Emission Depletion (gSTED) Microscopy

Samples with DC-SIGN antibody capture were prepared as described above. Captured DC-SIGN antibody was stained with goat anti-mouse Alexa Fluor 594. ATG9 was stained using indirect immunofluorescence with Abberior Star 635P secondary antibody. gSTED microscopy with time-gated detection was performed on a Leica SP8 TCS STED microscope (Leica Camera, Wetzlar, Germany) equipped with a pulsed white light excitation laser (NKT Photonics, Birkerød, Denmark). Dual-channel STED imaging was performed by sequentially exciting Abberior Star 635P and Alexa 594 at 646 nm and 598 nm, respectively. Both dyes were depleted with a 775 nm STED laser. A single optical section was acquired with an HC PL APO CS2 100×/1.40-N.A. oil objective (Leica Camera, Wetzlar, Germany), a scanning format of 1024 × 1024 pixel, 8-bit sampling, and 6-fold zoom, yielding a pixel dimension of 18.9 × 18.9 nm. Time-gated detection was set from 0.3–6 ns for all dyes. Fluorescence signals were detected sequentially by hybrid detectors at the appropriate spectral regions separated from the STED laser.

### 4.14. Statistical Analyses

The One-way ANOVA unpaired analysis was applied for statistics from data regarding DC-SIGN co-localization with ATG-related factors in Figure 3. Two-tailed Student’s *t*-test for paired observations (differences in the kinetics of receptor engagement compared to 0min for each donor or cell type) or unpaired parameters (Differences between HEK293T transfectants pharmacologically treated or not). Statistical analyses were performed using GraphPad Prism 6.0 (GraphPad software, San Diego, CA, USA) and significance was set at *p* < 0.05 (* *p* < 0.05; ** *p* < 0.01; *** *p* < 0.005; **** = *p* < 0.0001).

## Figures and Tables

**Figure 1 ijms-24-09008-f001:**
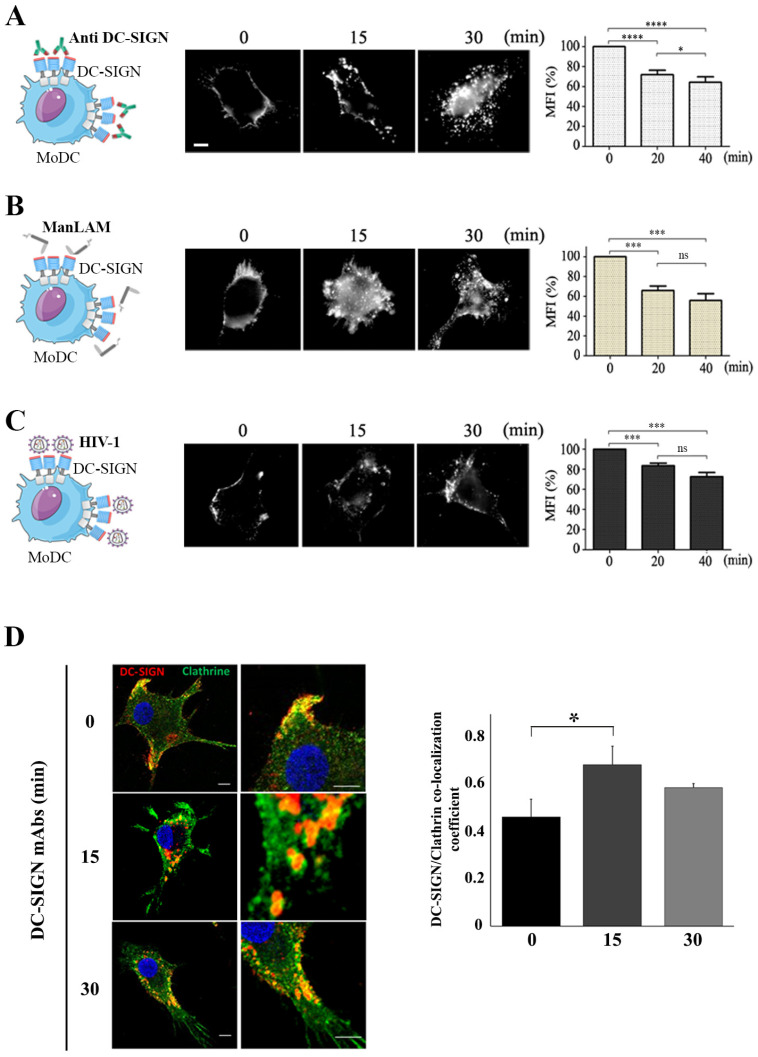
Ligand-mediated DC-SIGN endocytosis. Kinetics of DC-SIGN endocytosis measured by immunofluorescence (**middle** panels) or flow cytometry (**right** graphs) in primary MoDC treated with (**A**) specific DC-SIGN mAbs (5 μg/mL) (n = 7), (**B**) ManLAM (2 μg/mL) (n = 6), or (**C**) HIV-1-R5 (MOI of 2) (n = 4). Scale Bars in inset imaging correspond to 5 μm (**D**) Representative experiment showing co-localization between DC-SIGN (red) and clathrin (green) upon treatment of MoDC with DC-SIGN mAbs as above. Quantifications of co-localization events were performed using ImageJ and Manders co-localization coefficient represented in the histogram graph (n = 3). Scale bars correspond to 5 μm. Statistical significance are ns = not significant; * = *p* < 0.05; *** = *p* < 0.005; **** = *p* < 0.0001.

**Figure 2 ijms-24-09008-f002:**
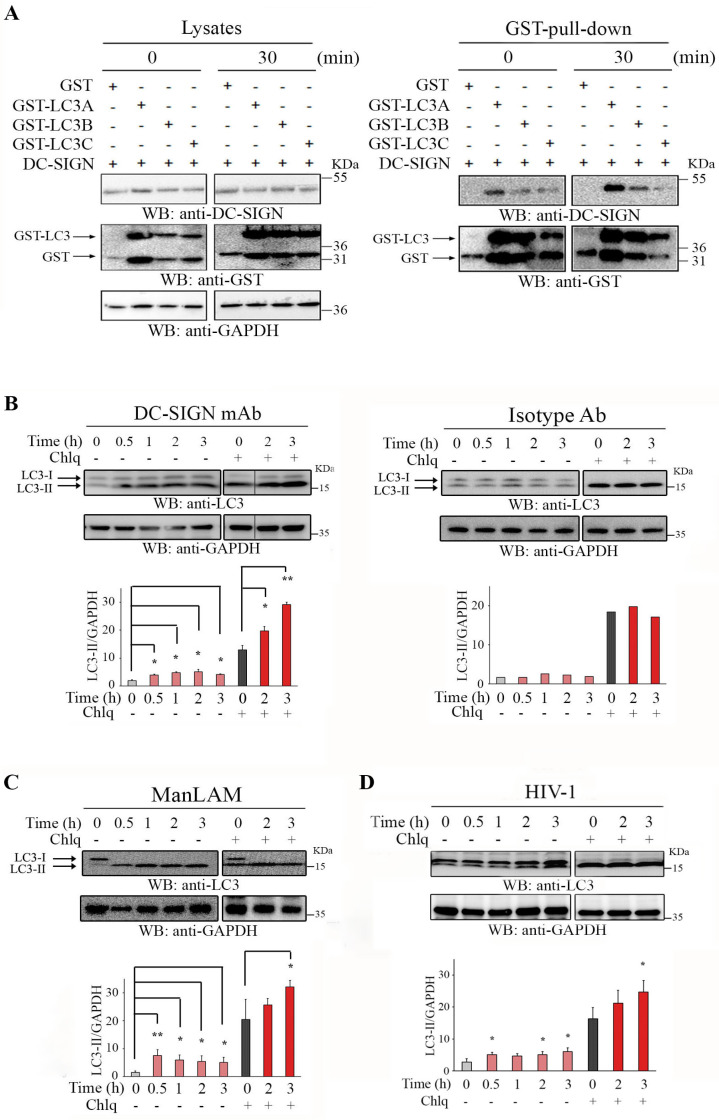
DC-SIGN engagement induces autophagy and associates with LC3. (**A**) HEK293T cells were co-transfected with plasmids expressing DC-SIGN WT and GST-expressing constructs as indicated. Cells were engaged or not with DC-SIGN mAbs (5 μg/mL) for 30 min before pull-down. Lysates (input control) and GST pull-down were immunoblotted with anti-DC-SIGN (**upper** panels), anti-GST (**middle** panels), and anti-GAPDH for input control. This experiment is representative of 4. (**B**) Lysates from MoDC treated for indicated times with DC-SIGN specific mAbs were immunoblotted with anti-LC3, and loading was controlled with anti-GAPDH (**upper** blots). Negative control experiments were performed using lysates of cells incubated with non-specific mouse serum (Isotype Ab) and immunoblotted as before (**lower** blots). Pre-treatments of MoDC with chloroquine (50 μM) were performed to validate functional autophagy flux as evidenced by LC3-II increase. Densitometry values and LC3-II/GAPDH ratio obtained from lysates of primary MoDC treated with DC-SIGN mAbs (n = 6) or Isotype mouse Abs (n = 3) were obtained for each time point of each condition and graphically reported on the right. The same experiment as above was performed but with lysates from MoDC treated with (**C**) ManLAM (2 μg/mL) (n = 4) or (**D**) challenged with HIV-1-R5 (MOI of 2) (n = 3) for indicated times. LC3-II/GAPDH ratio from densitometry analyses was graphically represented as before. Statistical significance: * = *p* < 0.05; ** = *p* < 0.01.

**Figure 3 ijms-24-09008-f003:**
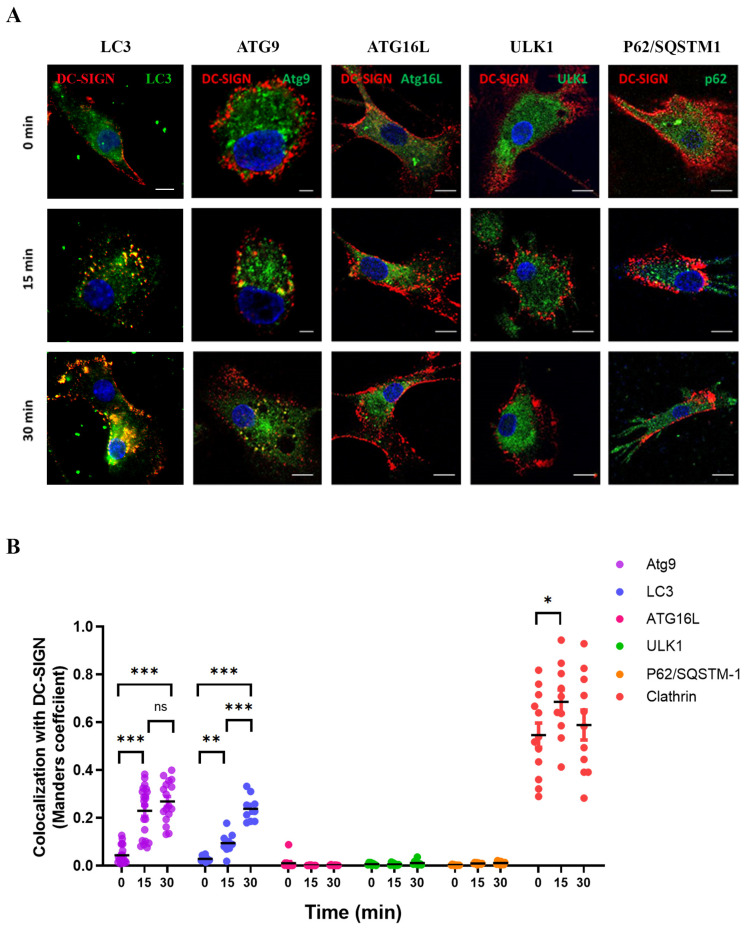
Imaging analysis of DC-SIGN endocytosis associated with ATG-related proteins. (**A**) Primary MoDC previously coated on coverslips were treated with DC-SIGN specific mAbs (5 μg/mL) for the indicated times. Cells were fixed, washed, and stained to detect DC-SIGN (red) and ATG-related proteins LC3, ATG9, ATG16L, ULK1, and P62/SQSTM-1 (in green) as displayed. The experiment shown is representative of 3 independent donors (n = 3). Scale Bars = 5 μm. (**B**) Images were treated with ImageJ to measure DC-SIGN/ATG9 co-localization coefficients from 30 cells/condition/donor. The obtained co-localization coefficients were graphically represented. Co-localization with clathrin was also performed as a control. Statistical significance: ns = not significant; * = *p* < 0.05; ** = *p* < 0.01; *** = *p* < 0.005.

**Figure 4 ijms-24-09008-f004:**
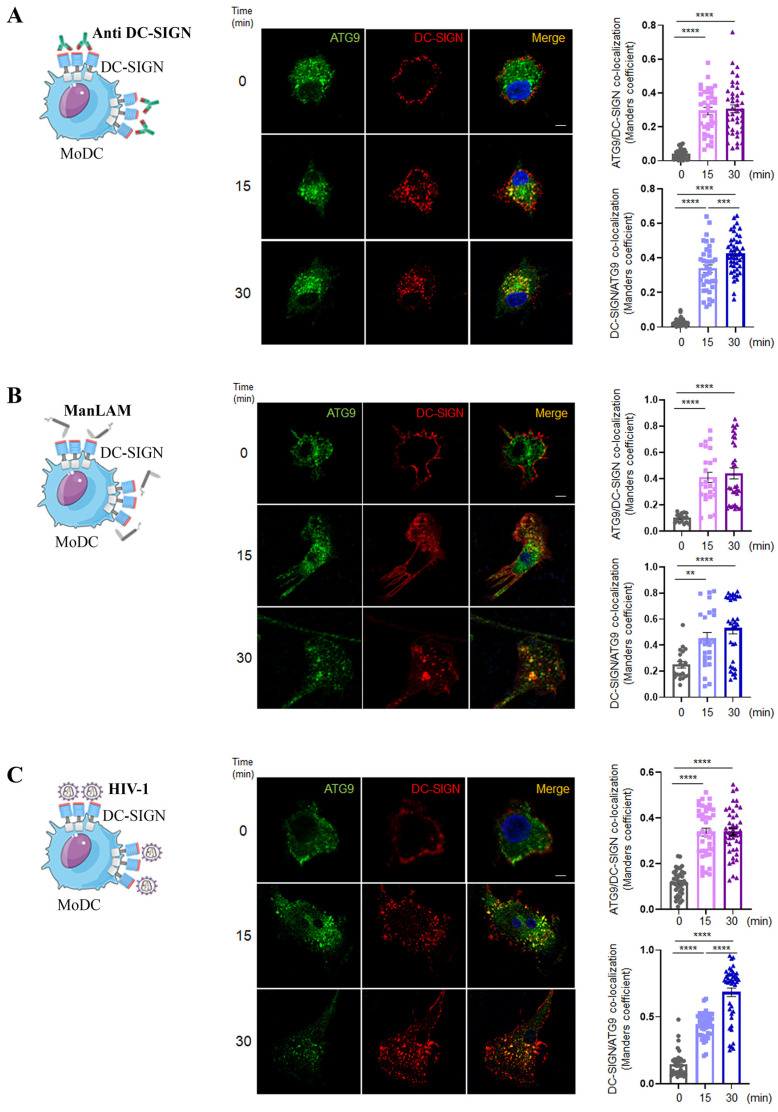
ATG9 co-localizes with endocytosed DC-SIGN. (**A**) Primary MoDC previously coated on coverslips were treated with DC-SIGN mAbs (5 μg/mL) (upper panels) for the indicated times. Cells were fixed, washed, and stained to detect DC-SIGN (red) and ATG9 (green). The white bar length in panels corresponds to 5 μm. The same as above was conducted with cells incubated with (**B**) ManLAM (2 μg/mL) or (**C**) HIV-1-R5 (MOI of 2). For each condition, images were treated with ImageJ to measure DC-SIGN/ATG9 co-localization coefficients from at least 30 cells/condition for each donor. The obtained DC-SIGN/ATG9 co-localization coefficients were graphically represented for conditions with DC-SIGN mAbs (n = 8 donors) (Histograms in 4**A**), ManLAM (n = 6 donors) (Histograms in 4**B**), and HIV-1-R5 (n = 5 donors) (Histograms in 4**C**). Statistical significance: ** = *p* < 0.01; *** = *p* < 0.005; **** = *p* < 0.0001.

**Figure 5 ijms-24-09008-f005:**
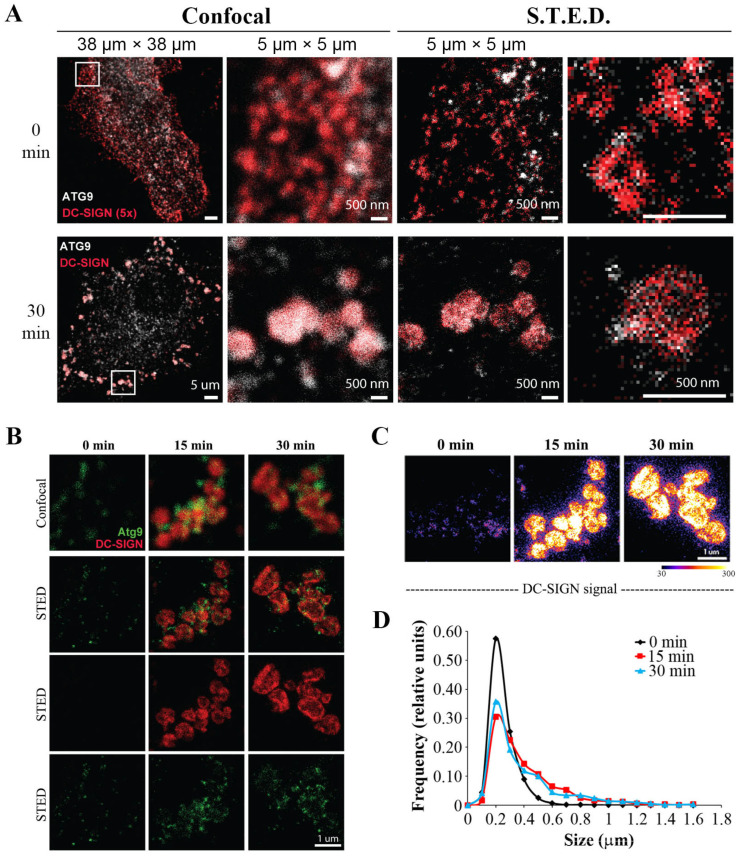
ATG9 forms nanoclusters with endocytosed DC-SIGN. (**A**) Primary MoDC were treated with DC-SIGN mAbs for the indicated time. Cells were fixed, washed, and stained to detect DC-SIGN (red) and ATG9 (grey). The same fields were analyzed by confocal (left) or STED (right) microscopy parameters. Thirty min conditions were taken with five times less excitation to avoid saturation. Shown pictures are representative regions from 6 to 10 cells. The mentioned resolution scale above each panel corresponds to the square displaying each image. (**B**) The same STED analysis as above was conducted on primary MoDC treated with DC-SIGN mAbs for indicated times. (**C**) A fire color scale was used to highlight differences in intracellular DC-SIGN labeling from the above samples. (**D**) Graph representing the Feret’s diameter of intracellular DC-SIGN positive structures upon kinetic receptor engagement (n = 6–10 cropped areas).

**Figure 6 ijms-24-09008-f006:**
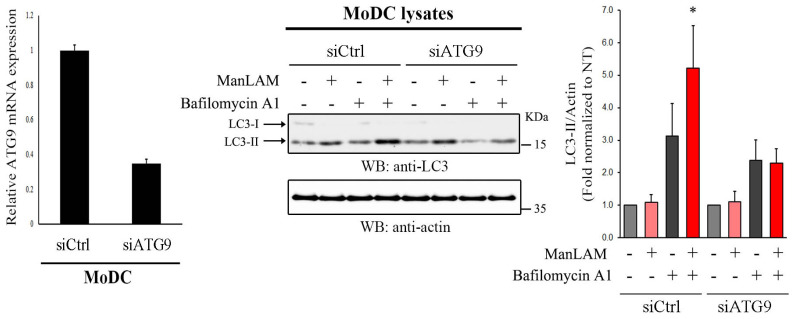
ATG9 is required for DC-SIGN-mediated autophagy flux activation. Primary MoDC were treated with irrelevant siRNA (siCtrl) or siRNA against ATG9 (siATG9), and ATG9 expression was controlled by RT-qPCR (**left** graph). Lysates from MoDC transfected as above and pre-treated with bafilomycin A1 (50 nM) for 1 h before stimulation with ManLAM (2 μg/mL) for 2 h were immunoblotted with anti-LC3 (**upper** blot). The loading control was performed with anti-actin (**lower** blot). LC3-II/actin ratio from densitometry analyses obtained from 3 independent experiments (n = 3) was normalized to untreated controls of each siRNA condition and graphically represented (**right** graph). Statistical significance: * = *p* < 0.05.

**Figure 7 ijms-24-09008-f007:**
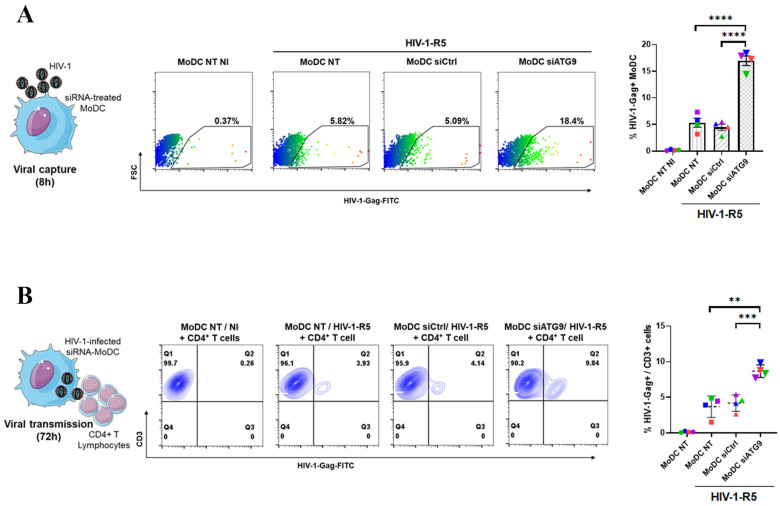
ATG9 expression lowers HIV-1-R5 viral input and limits DC-mediated transfer of infection towards CD4^+^ T lymphocytes. (**A**) Primary MoDC were left untreated (NT) or treated with irrelevant siRNA (siCtrl) or siRNA against ATG9 (siATG9) and challenged with HIV-1-R5 for 8 h when indicated. Cells were then fixed, washed, and permeabilized before staining with HIV-Gag Abs and analyzed by flow cytometry. The acquisition was set to 5.000 viable cells. A representative dot-plot experiment out of 4 is displayed, and a graph showing the % of HIV-Gag^+^ MoDC is presented (**right**). (**B**) MoDC treated as above were co-cultured for 72 h with autologous CD4^+^ T cells. Cells were then fixed, washed, permeabilized, and stained with HIV-Gag Abs (Kc57-FITC) and anti-CD3-APC before analysis by flow cytometry. The acquisition was set to 5.000 viable cells. A representative dot-plot experiment out of 4 is displayed, and a graph showing the % of HIV-Gag^+^/CD3^+^ cells is presented (**right**). Statistical significance: ** = *p* < 0.01; *** = *p* < 0.005; **** = *p* < 0.0001.

**Figure 8 ijms-24-09008-f008:**
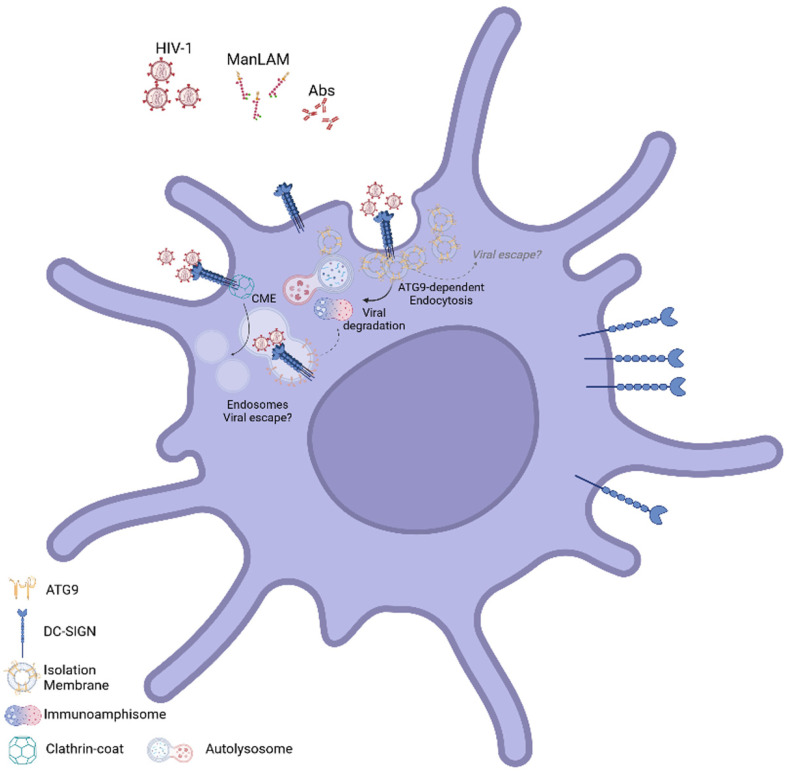
Graphical summary. ATG9-containing isolation membranes associate with the receptor DC-SIGN upon engagement by different ligands (Abs, ManLAM, and HIV-1-R5). Receptor engagement generates membrane proximal ATG9/DC-SIGN nanoclusters for which a pool might traffic through the endolysosomal pathway, thus limiting viral input and DC-mediated transfer of infection.

## Data Availability

When applicable, data from this study can be provided upon reasonable request.

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
