# Peer review of "The Autophagy Nucleation Factor ATG9 Forms Nanoclusters with the HIV-1 Receptor DC-SIGN and Regulates Early Antiviral Autophagy in Human Dendritic Cells"

_ijms, 2023, doi:10.3390/ijms24109008_

Round 1

Reviewer 1 Report

This is a very well-written and interesting paper about a non-canonic endosomal pathway, DC-SIGN-ATG, which regulates DC-mediated transmission of HIV-1 infection to CD4+ lymphocytes followed by induction of host antiviral response. The concept is well-articulated, and the data look convincing. However, there are certain things that can be further clarified.

1.    The title of the paper is a bit ambitious, and it is not clear from the title whether triggering of the receptor activates or inhibits autophagy. to remove or destroy HIV.

2.    It should also be clear from the abstract whether HIV can be degraded or removed by this pathway.

3.    There are no data supporting the downstream part of autophagy, which is lysosomal degradation since ATG9 by itself cannot degrade HIV. The level of HIV gag proteins is shown to prove that this pathway is related to autophagy-mediated degradation; however, the studies on other cell types like hepatocytes showed that p24 can be degraded not only by lysosome but by proteasome as well since it was stabilized by proteasome inhibitors (doi: 10.3390/biom9120851). Another study on HIV-infected hepatocytes demonstrated that lysosomal biogenesis may be suppressed by HIV (doi: 10.3390/biom11101497). Of course, the effect on lysosomes may be cell-specific, but still, lysosomal degradation as a downstream part of the pathway should be addressed/discussed.  

4.4.    A schematic diagram that connects all the findings will be helpful.

Author Response

We would like to thank the reviewer for his/her pertinent and justified comments.

Below are our replies in a point-by-point format:

REVIEWER 1

  1. The title of the paper is a bit ambitious, and it is not clear from the title whether triggering of the receptor activates or inhibits autophagy. to remove or destroy HIV.

As shown in Figures 2, 6, S1, S3 and S4, engagement of the receptor induces an early and potent autophagy flux in human MoDC, reminiscent of the early autophagy-dependent antiviral response previously described by us and others in myeloid cells (Blanchet et al., 2010; Fletcher et al., 2018; Orvedahl et al., 2011). Nevertheless, we agree with reviewer that the title of our study should rather bring attention to ATG9 and we have therefore proposed a new title without any change in the main message: "The autophagy nucleation factor ATG9 forms nanoclusters with the HIV-1 receptor DC-SIGN and regulates early antiviral autophagy in human Dendritic cells"

  1. It should also be clear from the abstract whether HIV can be degraded or removed by this pathway.

We’d like to thank the reviewer for this suggestion and we have now slightly modified the abstract by adding the following sentence: "Finally, Stimulated emission depletion (STED) microscopy performed in primary human MoDC revealed DC-SIGN-dependent submembrane nanoclusters formed with ATG9 which was required to degrade incoming viruses and further limit DC-mediated transmission of HIV-1 infection to CD4+ T lymphocytes."

  1. There are no data supporting the downstream part of autophagy, which is lysosomal degradation since ATG9 by itself cannot degrade HIV. The level of HIV gag proteins is shown to prove that this pathway is related to autophagy-mediated degradation; however, the studies on other cell types like hepatocytes showed that p24 can be degraded not only by lysosome but by proteasome as well since it was stabilized by proteasome inhibitors (doi: 10.3390/biom9120851). Another study on HIV-infected hepatocytes demonstrated that lysosomal biogenesis may be suppressed by HIV (doi: 10.3390/biom11101497). Of course, the effect on lysosomes may be cell-specific, but still, lysosomal degradation as a downstream part of the pathway should be addressed/discussed.

We agree with reviewer that cellular mechanisms involved in viral degradation might rely on more than one cellular degradation machinery. The proteasome was indeed reported to act early upon HIV-1 entry (Schwartz et al., 1998; Butler et al., 2002; Wei et al., 2005; Groschel & Bushman, 2005; Santoni de Sio et al., 2006), although this was observed in immortalized epithelial and CD4+ lymphoid cell lines While the Ubiquitin-Proteasome System (UPS) was reported to limit HIV-1 infection and be hijacked by some viral proteins to degrade cellular counterparts, a very early and direct effect of proteasome on HIV-1 incoming virions was however called into question (Duek et al., 2007; Rojas et al., 2019).  As such, while examples provided by the reviewer could be also relevant in human myeloid cells, the HIV-1-R5 infection conditions appear significantly different from our HIV-1 capture assays. Indeed, in the articles from Ganesan et al. (Biomolecules, 2019) and New-Aaron et al. (Biomolecules, 2021) (both manuscripts from the same group), hepatocytes were infected for 3 days thus leaving time for the cell to set and activate paralleled degradation pathways. In our model, however, we mainly focus on early events of HIV-1 infection of MoDC and thus analyse viral capture upon few hours of HIV-1 challenge. In these settings, we and others have reported that the lysosomal degradation pathway represent the main cellular catabolic pathway involved in the degradation of 80 to 90% of the viral input (Taccheti et al., 1997; Moris et al., 2004; Moris et al., 2006; Turville et al., 2004) although a substantial amount of viral input could escape lysosomal-mediated degradation and possibly fuel a so-called Virus-Containing Compartment (VCC) (Garcia et al., 2005; Turville et al., 2004). Our model is therefore closer to the suggested involvement of early lysosomal-mediated degradation pathway and we provide here evidence that ATG9 plays a critical role in its initiation. This again does not preclude the involvement of a proteasome-dependent pathway but will need further investigation to clarify this.

As suggested by reviewer, we have now added a short comment on this aspect in the discussion section: line 371 "The association with ATG9+ isolation membranes seems therefore to initiate a very early trafficking fate of incoming virions towards the endo-lysosomal degradation pathway as previously suggested."

4.4.    A schematic diagram that connects all the findings will be helpful.

A schematic diagram was indeed added as a supplemental figure (Fig. Sup 5) but we have now switched a modified version of this schematic figure in the main manuscript as requested by reviewer and therefore added a graphical summary as Figure 8.

Author Response

We would like to thank the reviewer for his/her pertinent and justified comments.

Below are our replies in a point-by-point format:

REVIEWER 2

Major comments:

The paper fits the topic of the special journal issue focusing on Cellular and Viral Immunology of
HIV-1 Infection: An Update. The novelty of this paper is in regard to its description of a DC-SIGN
ligation mediated-autophagy that may be implicated in decreasing HIV viral infectivity and cell to
cell transmission. The paper is well written, and the figures are clearly presented.
In general, the results seemed reproducible and matched the authors conclusions, however the
following points needs to be explained:

1- In Materials and Methods section, the authors describe that “HEK293T cells were co-
transfected with plasmids expressing DC-SIGN WT and either plasmids expressing GST-
fusion proteins or GST alone”, line 539. However, in Figure 2A, we can clearly see 2 bands
corresponding to GST molecular weight in lysate of cells transfected with GST-fusion proteins
that in this case are expected to show only the band around 45 KDa. (In the reference 70, a faint band of GST was also reported in GST-LC3 fusion transfected cells but not as intense as the bands in the current figure)

As correctly raised by reviewer, our immunoblots on lysates from transfected HEK cells show 2 bands in the GST-LC3 (A, B, C) transfection conditions. While the upper bands correspond indeed to the fused GST-LC3 entity, as confirmed by MW (~45-50 KDa), the lower bands from each lane (~32 KDa), also recognized by the specific anti-GST Abs, present a migration profile similar to the band appearing when GST alone is expressed. Therefore, the presence of GST alone in the GST-LC3 transfection conditions is clearly reminiscent of an intracellular cleavage occurring between the GST and LC3 protein sequence, a site usually targeted on purpose by proteolytic activity in vitro. For clarity, each entity has been marked by an arrow on side of the immunoblot panels in Figures.

2- Based on Figure 2B, authors show an increase of LC3-II following DC-SIGN ligation as reported by a denser band at 1h post-ligand addition compared to baseline. However, this result is in contradiction with that shown in supplemental Figure 3B, where in absence of Baf A1 the LC3-II is decreased compared to baseline.

We thank the reviewer for this remark. Indeed, the observation made by the reviewer is based on the dynamics of the modified form of LC3 analyzed in different cell types. Indeed, while Figure 2 is related to autophagy flux in human primary monocyte-derived DC, Figure Sup 3 concerns a DC-SIGN-expressing HEK-based cellular model. The only thing we can tell on such autophagy flux dynamics is that perhaps MoDc present a more potent flux, particularly when cells are engaged with agonists such as ManLAM and/or specific receptor Abs. However, a correct interpretation of a dynamic autophagic flux depends on the analysis of the LC3-II profile (increase or decrease) upon pre-treatment or not with a late-step autophagy flux inhibitor (Bafilomyin A1, Chloroquine…). In our settings, pre-treatment of MoDC and HEK cells with autophagy flux/lysosomal inhibitor evidenced an increase in LC3-II suggestive of autophagy flux induction.

3- Also, in Figure 2B, 2C and 2D, although using the same MoDC cells, the baseline expression of LC3-I seems different. Could the authors explain this result and address whether it can influence the levels of LC3-II. Additionally, the LC3-I bands appear to disappear after Chlq addition in isotype Ab graph (right) but not the DC-SIGN mAB (left).

We thank reviewer for this comment. Indeed, Western-blots from MoDC lysates shown in Figure 2 are not all from the same donors which might explain why basal LC3-I/LC3-II ratio are not showing the same pattern. While Figure 2B lysates were obtained from the same donor in order to have Isotype conditions as internal control for the DC-SIGN Ab-mediated stimulation, immunoblots in lysates shown in Figure 2C and Figure 2D were from different human blood donors. The purpose of this figure was not to compare between conditions but rather within each stimulation condition itself with the observed increase in LC3-II entity, particularly upon pre-treatment with Chlq (or BafA1 in other figures). Of note, these experiments were repeated at least 3 times for each stimulation condition.

4- In Figure 3B, was there any statistical difference between time 0 and 15 or 30min for DC- SIGN/LC3 colocalization data?

Upon reviewer’s comment, we have now added statistics in Figure 3B which were indeed missing in the submitted version. Statistics were obtained for individual autophagy-related factor comparing each time-point with a One-Way ANOVA unpaired test. Indeed, there is a statistical difference between time 0 and 15 or 30 min for DC-SIGN/LC3. Of note the difference appeared also significant between 15 and 30 min suggesting an accumulation over time of internalized DC-SIGN within LC3+ structures.

Minor comments:

  • Typos/spacing: line 103,198.

We have now corrected the spacing typos.

Round 2

Reviewer 2 Report

Thanks for the authors for answering the questions and including the statistical analysis.